# Review of Behavioral Psychology in Transition to Solar Photovoltaics for Low-Income Individuals

**Fransisca Angelica Rahardja [1], Shih-Chih Chen [2]** and **Untung Rahardja [1,\*]**

[1] Science and Technology Faculty, University of Raharja, Jenderal Sudirman Road No. 40, Tangerang 15117, Banten, Indonesia; fransisca@raharja.info
[2] Department of Information Management, National Kaohsiung University, 700, Kaohsiung University Rd., Nanzih District, Kaohsiung 811, Taiwan, China; scchen@nkust.edu.tw
\* Correspondence: untung@raharja.info

**Abstract:** The increase in nonrenewable energy (non-RE) has been a growing concern for low-income individuals' quality of life, health, economy, and environment. At the same time, the use of non-RE is also a great concern for the whole population as we are breathing the same environment. The photovoltaics (PV) solar panel is one solution to decrease low-income individuals' energy bills and increase the quality of life of all individuals. Knowing the behavioral theory of why low-income individuals do not adopt PV would allow further insights and possible interventions to help low-income individuals install PV. Research has found that low-income individuals are more likely to have financial and knowledge barriers that hinder them from installing PV. Providing a way for low-income individuals to combat these barriers would help them to use PV. This review showed that low-income individuals are likely to benefit from policy programs that incentivize them to use PV. More knowledge about PV can also be aided by policy programs that inform low-income individuals how to save financially and at the same time work their way to install PV. Social groups could also be formed in the same policy programs to help low-income individuals share strategies on saving financially and knowledge about the benefit of installing PV. These social groups can act as a social reinforcement to low-income individuals to install PV. Helping low-income individuals to install PV would help low-income individuals financially and improve the population's quality of life.

**Keywords:** renewable energy; photovoltaics; low-income individuals; environment; reinforcement

## 1. Introduction and Background

The transformation of low-income individuals to the usage of renewable energy (RE) such as photovoltaics (PV) solar panels is a major strategy to solve climate change. A lot of countries with high exposure to sunlight have now moved to use PV to decrease their energy usage and, at the same time, decrease energy bills [1]. However, although many people have known the benefits of PV, low-income individuals continue to face challenges in installing PV in their houses [2]. Non-RE energy usage of low-income individuals has been an issue for the low-income and non-low-income individuals alike. This is because non-RE usage has an adverse effect on the environment's air quality shared by all the Earth population. Knowing the barriers that hinder low-income individuals from adopting the use of PV is essential to increasing the health and environment of the whole population.

The increased use of nonrenewable energy (non-RE) has been an ongoing worldwide concern for many years. These non-RE byproducts, such as $CO_2$ emission, were shown to adversely impact the Earth's heat, air, and water quality [3,4]. The following non-RE byproducts have also been known to negatively impact all individuals' physical and psychological health [5,6]. Mikhaylov et al. [5] further iterate that the increased temperature due to the accumulation of greenhouse gases and the exhaust fumes of transportation is a major issue for continuously increased sea levels. This rising seawater will lead to most parts of the land being flooded, less harvest, and less land space for the increasing

population to create a shelter if not properly mitigated. With all of these adverse effects, it is of great importance that we find ways to decrease the use of non-RE.

While decreasing the use of non-RE, researchers in many different interdisciplinary studies have also been collaborating to increase the use of RE [7–11]. To its success, there are now many new innovative ways to decrease the use of non-RE. For example, solar photovoltaic (PV) extracts the light radiation energy from the sun to power up households and organizations electricity [12,13]. Due to technological advancement, the use of RE has continuously increased. Arıoğlu et al. [14] and IEA [13] data showed RE usage has risen from 12% to 29% from 2012 to 2020. The following statistics also showed that high-income countries such as China had installed RE such as a PV more than low-income countries [13].

Until today, PV is the most accessible type of RE installed by private entities and households for both high- and low-income individuals alike. The deployment of a smart grid coupled with advanced blockchain technologies accelerates the mass adoption of PV throughout the globe [15,16]. Because of this characteristic, PV has been the fastest-growing type of RE compared to other types of RE [17]. However, the use of PV is less seen in low-income countries because low-income countries have more low-income individuals [13]. Low-income individuals have different barriers compared to high-income individuals, causing them to struggle to install PV. According to National Renewable Energy Laboratory [18], both low-income and high-income individuals have similar personal obligations and personal interests to address environmental issues. However, although both groups have similar values, low-income individuals countries were seen to install PV less than high-income countries. Therefore, it is important to research the barriers of low-income individuals installing PV. Helping low-income individuals to install PV helps them financially and increase their quality of life [19].

Although technological advancements were made, many low-income individuals continuously faced barriers to adopt RE, such as PV. These difficulties are largely due to the financial difficulties that low-income individuals encounter. Berry et al. [20] and Drehobl and Ross [21] studies found low-income individuals spent most of their income on energy bills. Both of these studies also showed that low-income individuals use a higher percentage of their income for energy bills compared to higher-income countries. This is true even though low-income individuals use less electricity per square foot than higher-level income [21]. Treadway [22] survey data also mention that one-third of the population in the United States encounter financial difficulties to pay their bills several times in their lives. Some groups of low-income individuals are reported to forego food and health expenses to pay for their energy bills [20,23]. With these difficulties, even if investment in PV help low-income individuals financially in their energy burden, these group found great difficulties installing PV with upfront payment [2,24]. Energy burden is defined as the proportion of financial resources among low-income individuals allocated on energy expenditure [25].

The definition of low-income individuals would help us narrow down who can benefit from installation PV. How low-income individuals can benefit from PV installation would need to depend on the available resources in each country and state. Additionally, it also depends on the financial programs which help low-income individuals install PV [2]. There are different definitions of low-income individuals used in research and low-income funding eligibility programs. Some studies defined low-income individuals as having less than 60% less of the median asset value than the population [26]. Some other studies defined low income as individuals who have low skills in finding jobs, have limited human capital, have a lack of education, and live in areas of poverty [27–29].

There were other definitions of low income depending on the needs of the study. Some other definitions of low income include a person who cannot maintain sufficient warmth at a reasonable cost to keep themselves healthy, family income below 200% of the Federal poverty threshold, and others [30,31]. Additionally, there is also a different low-income definitions on different countries. In Finland, low income is defined as households who have income lower than 60% of the median money income of all households across the

country [32]. On the other hand, US low income is defined as four people in the households that have an income 50–80% of the state area [33].

Studies and different countries have different definitions of low income. With different definitions, it is difficult to analyze how many low-income individuals and in what circumstances can low-income individuals benefit from PV support programs. If multiple studies were to be analyzed with different definitions, there would be low validity in the research results indicating how many percentages of low income can or cannot benefit from the low-income RE support program. With this, research on what circumstances low-income definition can benefit from PV would warrant further research. The population that falls in very low income would possibly have the more pressing financial issue that hinders their ability to install PV further. However, it is essential to note that a great majority of low-income individuals can benefit financially with the help of government low-income PV support programs [2].

It is important to know the barriers that hinder low-income individuals from investing in PV to encourage them to install PV. Knowing what factors hinder low-income individuals from investing in PV can help us tailor the appropriate intervention to help low-income individuals invest in PV. Research has shown that low-income individuals are more likely to have difficulties saving money for emergencies [34]. They are also likely to neglect home maintenance [35] and forego profitable small investments to cover their immediate payments [36]. These financial difficulties that they encounter negatively impact their ability to make financial decisions and save for medical emergencies [34]. These studies suggest that low-income individuals have less room to make financial mistakes [37]. These difficulties are financial barriers that hinder low-income individuals from investing in PV.

Other factors other than financial difficulties that hinder low-income individuals from installing PV are not knowing the benefit of PV and how to install PV. These knowledge barriers hinder low-income individuals from installing PV even if they agree that PV would greatly help their financial difficulties in the long term [22]. It is important to highlight that agreeing to install PV does not necessarily mean individuals fully understand the benefit of PV and how to install it [22]. Brown et al. [2] also found some low-income people encounter difficulties finding financial support programs designed to help them install PV. Therefore, even if low-income individuals believe installing PV benefits them and financial support is in place, this lack of knowledge support and uncertainty about how to apply PV utilities hinders them from installing PV [2].

RE system implementation such as PV greatly benefits the low-income individual's expenses. Walton [8] and Samarripas and York [9] mention that investment in PV decreases energy bills, increases the quality of life, and significantly reduces energy costs for low-income individuals. Walton [8] also mentions that RE lead to a higher quality of life for low-income individuals beyond cost savings. Additionally, helping low-income individuals to invest in RE also improves the country's economy. As low-income individuals start to install PV, there is less of a burden for the governments to support low-income individuals financially. With less financial needs to be directed to low-income individuals government could focus their finances on other sectors to improve the countries [38]. Figures 1 and 2 summarized the interaction of low-income individuals not installing PV or installing PV to low income individuals situation, environment, and the population. With all these benefits mentioned above, it is important to review the barriers of low-income individuals and potential interventions to help low-income individuals install PV.

The review of Steg et al. [39] suggests internal factors such as a person's value to sustainability play a big factor in the adoption of RE. The author further elaborated four types of values: hedonistic, egoistic, altruistic, and biospheric. Hedonistic refers to values focusing on comfort and pleasure. Egoistic refers to people valuing resources such as money or status. Altruistic refers to the value of the health and well-being of other people. Biospheric refers to values focusing on the consequences of nature and the environment. Steg et al.'s [39] study argued that people with biospheric values are more likely to have the intrinsic motivation to use PV as it aligns with their moral values than people rated

to be high in other values. This study further suggests that providing PV knowledge and aligning the knowledge with individual central values encourage them to install use PV. This study also suggests we are more likely to have a higher probability of encouraging individuals to use PV if the benefits that they attain are aligned with their values.

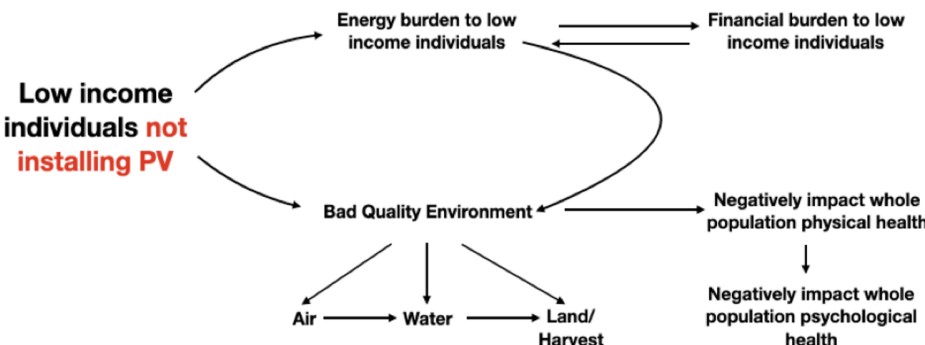

**Figure 1.** The interaction of low-income individuals not installing PV with their own financial situation. This figure also showed the interaction between low-income individuals not installing PV with the environment and the whole population health regardless of their income status.

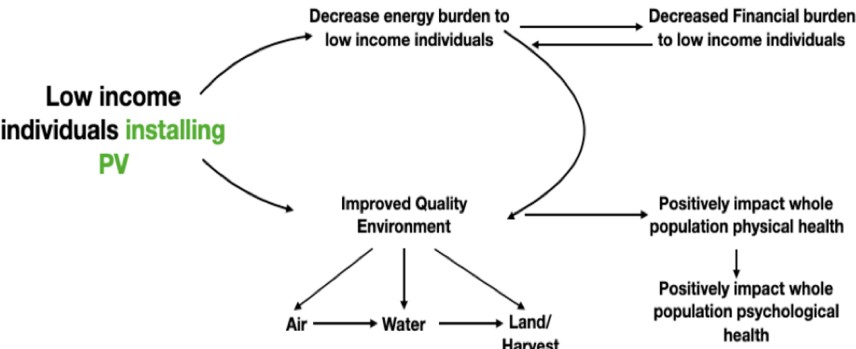

**Figure 2.** The interaction of low-income individuals installing PV with their own financial situation. This figure also showed the interaction between low-income individuals installing PV with the environment and the whole population health regardless of their income status.

Another internal factor to consider is the decision of multiple people in the households rather than just a single individual's decision to invest in PV. In many circumstances, the decision to invest in one household needs to be dependent upon the decision of multiple individuals [40]. Knowing how multiple people in the households interact based on their different traits is important to predict whether they are going to invest in PV. This research looked at surveys concerning the behavior of investing PV for individual people and how people in a household with different traits interact with each other. This research suggests that the interaction of individuals in multiple households with the different traits predictor affects their decision to invest in PV. Poier [40] research support Steg et al.'s [39] research suggesting that aligning individual and multiple people values in households leads to a higher probability of them installing PV.

Other factors play a critical part in the adoption of RE. Schultz et al. [41] and Nolan et al. [42] household energy survey participants reported other people's such as their neighbor's behavior as the least aspect that made them change their sustainability behavior. However, the results of the experiments showed the opposite results. Both Schultz et al.'s [41] and Nolan et al.'s [42] experiments showed that neighbors' sustainable behaviors are a high predictor of other people's behavior to do the same behavior. These studies suggest that social influence does affect people's behavior. Cialdini [43] suggests social impact affects behavior because people are constantly looking for social behavior

to indicate appropriate behavior in a given situation. This research suggests that as we intervene in groups of people with similar barriers towards RE, we are also encouraging other people nearby to follow and install PV.

These past research theories play a significant factor in helping people adopt PV. It provides insights into individuals' barriers to installing PV in their households. Additionally, these past studies also comprehensively mention the public psychological barriers and interventions to adopt PV. However, these study analyses did not include psychological barriers and interventions that focus on helping low-income individuals. Thus, these proposed techniques are unlikely to work with low-income individuals as low-income individuals are likely to have more barriers than non-low-income individuals. Figuring out why providing incentives increases PV adoption at the psychological level provides possible intervention to help low-income individuals install PV. Additionally, low-income individuals installing PV also leads to health and environmental benefits at the global level.

This paper aims to use behavioral psychology accompanied by other sectors of psychology to understand the psychological barriers of using PV in low-income individuals. We further propose a possible intervention to be tested to help low-income individuals with energy burdens. We divided each behavioral psychology theory into parts. Each part concentrates on a different behavioral psychology theory that explains low-income behavior to install PV. Related to the barriers of low-income individuals and our proposed behavioral psychology theory, we further propose a possible intervention to be tested.

## 2. Method

Low-income individuals' barriers are reviewed across multiple research papers in the introduction. These barriers serve as an important basis for the behavioral theory to predict behaviors. We further determine that the behavioral psychology theory aligns to predict the behavior of low-income individuals. These perspectives are important to predict low-income individuals' behavior in the presence of different policy programs. Additionally, these theories also uncover the problem that hinders low-income individuals from installing PV. These insights are then used to create possible interventions to help low-income individuals to invest in PV.

## 3. Defining Behavioral Psychology

Behavioral psychology is a framework of psychology that operates based on the assumption that organism behaviors are predictive based on their biology and environment [44]. Based on Baer et al. [44] study, behavioral psychology research focuses on observable scientific behaviors that can be quantitatively measured and avoid making assumptions that are not based on observable actions. To confirm the validity of the findings, the behavioral psychology relies on good design practices and various studies being replicated across different subjects, participants, settings, and times. Behavioral psychology studies first test the theory on the animal before the intervention is systematically replicated in humans [45]. Only if the study in humans experiment confirms the previous hypothesis that is performed in animals and the study benefits humanity, the intervention can then be used in the applied human settings. Behavioral psychology seeks to find out the fundamental of why humans behave in certain ways to figure ways to transfer it into more long-lasting beneficial behavior across different settings. Behavioral psychology might have a different philosophy, but it does not negate other parts of psychology research such as social, personality, and other types of psychology that affect human lives [46–50]. This section may be divided by subheadings. It should provide a concise and precise description of the experimental results, their interpretation, and the experimental conclusions that can be drawn.

### 3.1. PV Challenges in Low-Income Individuals—Matching Law Theory

The barriers to transition to RE differ from whether the citizens have low- and high-income individuals. Low-income individuals are less likely to fulfill their daily needs and

therefore are more likely to choose a system that is more likely to be less costly [51]. When low income leads to difficulty to fulfill daily needs, changing behaviors that lead to more costly finances might not be an option, even if they have sufficient knowledge to install PV [52]. In this case, what matters for the low-income individuals is how they can fulfill their biological needs to survive and minimize any necessary cost to ensure survival. In this regard, low-income individuals benefit from obtaining government incentives to install PV [53].

The basic principle of the matching law explains how incentives can be an important predictor of low-income individuals installing PV in their households. Based on the matching law, the behaviors allocated between the choices depend on how many reinforcers each alternative has [54]. If there are more reinforcers in one alternative than the other, the organism allocates more of its time in that particular choice than the other choice [55]. Based on their previous experience, organisms predict where the reinforcers are and allocate more behaviors to where they predict the reinforcers are [54]. It is important to consider that human reinforcer can be many different things such as financial, social, and other things [56,57].

Another dimension relevant to understanding how likely low-income individuals are to adopt PV is the sensitivity and biased parameter of the matching law theory [58,59]. The more reinforcing the rewards, the more sensitive the organism choices are toward that particular alternative [60–62]. Alternatively, biased refers to the tendency of the organism to prefer responding to one alternative over the other [58]. Other independent variables that affect the sensitivity and bias are how hard it is to obtain a reinforcer and the delays and magnitude of the reinforcer [58,63]. This research suggests that reinforcers such as financial incentives lead low-income individuals to be more sensitive toward the choice of installing PV. Thus, this meant that government incentives would increase the probability of behavior of installing PV in low-income individuals [64].

All of the basic behavioral principle settings provide an important perspective for the importance of government incentives to install PV. However, it is important to note that in applied settings there are more variables that can effect low-income individuals to install PV. Knowledge is shown to be one of the barriers of low-income individuals to adopt PV [2,65]. Without knowing the benefit, they would not see PV installation as something that is reinforcing for them. Thus, providing an incentive on its own would prove to be a redundant strategy for low-income individuals to change their course of behavior toward RE. Research suggests providing reinforcer in addition to knowing what can be done to decrease environmental problems and the benefit of doing it are effective to encourage people to improve their environmental behavior [60–62,66]. Behavior that only targets the knowledge of the importance of PV alone is less likely to be an effective strategy to transition individual to install PV [67].

Matching law theory can also apply in environmental public policy. The government and policymakers have constantly been discussing how to balance the environment and the need for a strong economy [68–70]. Although investing in PV is good for the environment, low-income consumers are less likely to invest in PV if the cost of investing in PV is too expensive or not affordable [68]. In relation to that, if the consumers did not invest in RE such as PV, the environment is not protected. With this, it is very important to provide the importance of investing in PV knowledge and government financial subsidy to convince the low-income consumers to invest in PV [64,66,68].

Palage et al.'s [68] study suggests that the government and public policy can encourage low-income individuals to invest in PV by providing financial subsidies to PV industry production, innovation technological advancement, and buyers. More supply and demand can be increased by providing financial subsidies to industry production and buyers in these sectors. By initiating this system, the economy can increase through the tax system, and at the same time, the environment can be protected. Public policy can also support technological advancement in the target to increase efficiency and lower the production cost of the PV. Based on matching law theory, providing financial subsidies and incentives in this

area increase the sensitivity of the production sector to invest their time in PV. Additionally, it also increases the sensitivity of the low-income individuals to buy PV. Based on matching law theory, increasing the price sensitivity toward all of these sectors improves the quality of the environment and the economy.

### 3.2. Social and Uncertainty Barriers—Punishment Behavioral Literature

Alternatively, as reinforcement increase behaviors, doing something that felt punishing to individuals decrease the frequency of behaviors [71]. Rasummusen and Newland's [71] study showed that most individuals felt three times more punishing to lose compared to obtaining the same value of goods. This study sets a game in which participants are able to earn monetary reinforcers if they win a computer game. In the punished, there were punishments toward one alternative. This study found that participants made 3× less responses in the punished condition compared to the unpunished condition. This study suggests a cent lost is valued much more compared to a cent that is earned.

Rasmussen and Newland's [71] study has applied implications to the low-income adoption toward PV. Research has found that the thought of uncertainty often discourages individuals from investing in the PV as they are uncertain how to benefit from RE [72]. Low-income individuals are uncertain if they benefit or if they were instead going to lose from this investing PV. Additionally, the pressing financial pressure experienced by low-income individuals further discourages low-income individuals from investing in PV [37]. It is important to note that learning to install PV requires low-income individuals' time and effort to learn about the benefit of PV, how to install them, and the monetary values required to install the PV. Based on the punishment literature and social psychology research, these uncertain feelings act as punishment discriminative stimulus [71,73]. Discriminative stimulus acts as a signal for the individuals to predict whether reinforcement or punishment is forthcoming [74]. Based on the following prediction, humans behave accordingly to attain the reward and avoid the events that they take as aversive. Feelings of uncertainty as to whether low-income individuals can attain financial benefits by investing in PV and the uncertainty of not knowing how to learn to install PV hinder low-income individuals' decision to install PV.

One way to intervene on uncertainty about the knowledge is to provide knowledge on how they can benefit from PV [65]. This can be performed by the government making social programs specifically for low-income individuals where they can gather and learn about the benefit of PV and how to install PV. Jones et al.'s [75] research showed that low-income individuals are able to reduce their bills when they are provided education on how to pay off their financial loans. Jones et al. [75] suggest providing low-income knowledge to save by adopting PV would also help them install PV and help them decrease their financial bills related to energy used in the household. Flyers and internet ads tell them about the program to support these plans. Additionally, governments could set a lottery draw for all the low-income individuals who attend the programs. Vlaev et al.'s [76] study showed that lottery, such as financial incentives, increases individuals' probability of performing a certain behavior. All of these studies suggest that financial incentives such as lottery and setting governmental social programs to educate the benefit of PV would help low-income individuals invest in PV.

As more low-income individuals are more educated to find the relevant support for them to benefit from PV, people nearby them are also more likely to be informed about the benefit of PV. Cialdini's [43] research suggests that people modify and imitate other people's behavior for socially appropriate behavior in a particular cultural context. The more people invest in PV, the more likely PV is to be a signal of long-term reward for low-income individuals rather than a signal of uncertainty [73]. With this, the more people invest in PV, the more likely people nearby with similar situations are to invest in PV.

Another way we can help low-income individuals adopt PV is to provide positive social reinforcement. Cialdini and James [77] also suggest that reinforcing people's sustainable behavior with social cues such as smiles and compliments worked to help people

engage in sustainable behavior. Positive signals such as compliments, smiles, and nods from a social context indicate that they are doing the socially acceptable behavior [57,65]. One way to obtain low-income individuals to invest in PV is to train the educators who teach low-income individuals to provide social reinforcers. Behavioral skill training (BST) research has shown that educators were more confident to provide reinforcers in a more natural way than educators who were not trained [78]. Miles and Wilder's [79] studies also showed interventions were more efficient with educators trained to provide reinforcers than educators who were not trained.

The government can also provide reinforcers by online comments after low-income individuals install PV. For example, an electricity company owned by the government could remind their customer, "good job, you use lower electricity this month due to PV installation (Manning, 2009). For more ways to know the benefit of PV, click the link below". Positive feedback such as the one above helps to nudge low-income individuals to acknowledge the benefit of using PV to a greater level [80,81]. The more they acknowledge the use of PV, the more likely they are to recommend it to other people around them about the benefit of PV.

### 3.3. Delay Discounting Preventing People to Install RE Even If They Have the Money

Delay discounting theory can provide a psychological perspective to why low-income individuals are less likely to install PV. Based on delay discounting theory, the individual tends to choose more immediate but smaller reinforcers than larger but more delayed reinforcers [82]. Kirby and Maraković's [83] study tested delay discounting theory in participants. They asked participants to choose between a range of smaller amounts of money today or larger amounts of money over a longer period of time. Results showed that most participants chose smaller and more immediate rewards than larger but more delayed rewards.

The equation of delay discounted is counted as below [82,83]. '*V*' stands for delayed rewards, '*A*' stands for the amount of reward, *k* stands for discounting rate parameter, and '*D*' stands for delayed. The below equation shows how rewards are likely to lose their value through time for different individuals. The smallest immediate money is always ranged from 15 to 83 dollars. The larger, more delayed rewards vary from 9 dollar difference to 55 dollars difference from the smaller, more immediate rewards. Another thing that they vary is the days that they are going to be receiving the more delayed reward. This study found that the longer the delays, the more likely individuals are to choose the more immediate reward than the later ones. This study also shows that only 14% of the individuals prefer the larger and later rewards.

$$V = \frac{A}{1 + kD} \quad \text{(Delay Discounting Preventing)}$$

Currently, the only price of PV is the installation price and maintenance cost [84]. However, the individual still needs a certain amount of revenue to buy the installation price. The current installation of PV is still perceived to be an option with a more expensive cost than slowly paying off the usage of non-RE. One situational example would be paying gas to cook food is cheaper and more affordable than installing RE because they do not understand how to gain from PV installation financially. This investment in PV is perceived as a larger long-term reward. On the other hand, the constant use of non-RE is seen as a smaller and more immediate reward. People who were unsure about the benefit of RE are more prone to choose the small immediate reward than long-term rewards. It could be argued that because of their financial difficulties, delay discounting is likely to be steeper in low-income individuals compared to individuals with higher income.

As mentioned above, one way to intervene in this issue is to provide government incentives and policymakers providing social groups to inform low-income individuals about the benefit of PV. These strategies help nudge low-income individuals to gain longer and more delayed rewards compared to smaller and more immediate rewards. On top

of that, another predictive variable that could supplement the incentive intervention is to involve low-income individuals in the policy decision-making process. Huijts et al.'s [85] study showed that people are more likely to adopt energy system changes when they are involved in decision making than when they are not. The following study further stresses the importance of people being fully informed of the process from the beginning rather than only being informed of the policy changes after the experts settle the policy. Earle and Siegrist [86] suggest people are more likely to trust the policy changes when people's interests are taken into account. Getting people involved in policy change is more likely to lead to a policy change that aligns with low-income individuals' values and perspectives [39]. Furthermore, a policy change that aligns with low-income individuals' values increases the reinforcement for low-income individuals to support others in investing in PV [87].

*3.4. Generalization Leading to Increase Use of PV and Other Proenvironmental Behavior*

Social programs that help support low-income individuals to understand the benefit of PV are also likely to lead to the generalization of other positive proenvironmental behavior [43]. Generalization is the adoption of similar or related behavior from the learned behaviors [88]. Hanson [88] showed the more similar the behaviors are to the learned behavior, the more likely the individual is to adopt a similar behavior. In relation to PV, positive generalization referred to similar proenvironmental behaviors other than the installation of PV. As more low-income individuals know about other people with similar situations installing PV from social policy programs, they were more inclined to install PV and do other proenvironmental behaviors. This is because the more people conducting the installation for proenvironmental reasons, the more proenvironmental reasons are likely to be seen accepted as socially appropriate behavior [43].

Generalization theory suggests that as people engage in environmentally friendly behavior such as investing in PV, and they would also do other environmentally friendly behavior such as decreasing the use of electricity [88]. Generalization has been successfully shown in many applied research of environmental behaviors [89]. Generalizations are more likely to happen when individuals are trained to associate multiple discriminative stimuli to signal reinforcers rather than just one stimulus [88]. In relation to the installation of PV, on top of installing PV, we can train low-income individuals to associate other proenvironmental reasons to decrease energy burdens.

For generalization theory to work better, it is important to emphasize the benefit that low-income individuals can obtain from investing in PV in relation to the hedonistic, egoistic, altruistic, and biospheric values elaborated by Steg et al. [39]. The following research suggests values act as an intrinsic motivation for people to invest in PV. Additionally, generalization theory suggests that the following values can act as a discriminative stimulus for people to perform similar pro-environmental behaviors that align with their values [88,89]. These studies suggest that to increase the probability of positive generalization toward the environment, we should target their intrinsic motivation to be the discriminative stimulus rather than investing in PV as discriminative stimuli for reinforcement.

One way to make low-income individuals' values as a discriminative stimulus to act proenvironmentally is to state the benefits of investing in PV and the other benefits of the environment. Research on stimulus relation suggests that multiple discriminative stimuli that evoke the same signal are likely to result in the same response [90]. The following study also suggests that we can train individuals to associate different discriminative stimuli to obtain the same response. It might be beneficial to compare the two contrasting discriminative stimuli to understand this concept. Suppose educators mention, "one benefit to invest in PV is we can save energy bills". By mentioning just the benefit PV, people are more likely to just associate PV as discriminative to save the energy bills rather than the personal values as a discriminative stimulus to save energy bills. Thus, people were more likely to only invest in PV and not engage in other environmentally friendly behavior. On the other hand, we can also associate the four values mentioned by Steg et al. [39]

with the benefits of PV. For example, educators can mention the following to low-income individuals to target individuals who have hedonistic values to invest in PV: "Just like how we decrease the use of electricity for other comforting reason such like buying food that we like, we can save energy bills by investing in PV to do the same thing". By aligning their values and PV benefits, low-income individuals were more likely to align their values as a discriminative stimulus to attain intrinsic reinforcers and do other proenvironmental behaviors similar to investing PV [90].

## 4. Result and Discussion

This paper discusses behavioral theory that explain low-income individuals behaviour to install PV. We propose four behavioral psychology theories for us to understand the behaviors of low-income behavior. First, with the theory of matching law, we argued individuals are more likely to adopt RE when they find the reinforcer in doing it. Second, with the research on punishment, we argued individuals are less likely to install RE when individuals are uncertain how they are going to benefit from the installation of RE. Third, with delay discounting theory, we argued most individuals would discount the value of a more rewarding but delayed reward. In low-income individuals, delay discounting is likely to be steeper as they would have less money to invest in PV. Lastly, with generalization theory, we argued there would be a similar proenvironmental behavior that low-income individuals engage in after investing in PV. This generalization theory can be supported by educators associating words that were aligned with by Steg et al.'s [39] hedonistic, egoistic, altruistic, and biospheric values.

The intervention that we proposed includes government intervention to help low-income individuals to invest in PV. Social PV groups would help low-income individuals increase their knowledge to save money in relation to PV use. This same group can also act as a social platform that low-income individuals can share their stories in their success and failure related to investing PV. To encourage people to attend the social platform designed to help low-income individuals install PV, positive reinforcement such as the possibility of winning the lottery for the people who attend the social groups can be implemented. Furthermore, we also encourage policymakers to involve low-income individuals when making a program to help low-income individuals adopt PV. These factors would increase the chance of low-income individuals to invest in PV. Table 1 summarizes each behavioral theory, problem uncovered based on past research, predictive behavior, and intervention used within this research.

It is worth noting that this financial and program support is already in place in some states and countries. Every country and every state has different programs that support low-income people financially on their energy cost burden [2]. Some programs such as 'Energy Efficiency and Conservation Loan Program' [91] are designed to provide low-income individuals loans to install PV. This decreases the upfront PV investment payment barriers that hinder low-income individuals from investing in RE. Other programs such as 'Solar for all' programs in the USA help to incentivize solar panels installed to help low-income individuals install PV [92]. Australia has also created 'peer-to-peer electricity' to incentivize people to install PV and save electricity [93]. These systems incentivized people to use less electricity and market the access electricity to other people in need. This peer-to-peer electricity system incentivized people to invest more in PV and use less electricity. However, all of these PV investment programs and incentives were seen less in low-income countries.

Another way that countries can improve the system would be to initiate higher taxes for higher-income individuals using non-RE. The following tax money could then be used for the government to educate the low-income individuals on the benefits, how to install PV, and how they can benefit in the long term. As individuals are sensitive to price, higher-income individuals are going to use less non-RE with the tax increase. Additionally, the following tax money can be used to subsidize technological PV advancement, PV suppliers, and incentivized PV buyers [68]. Solar for all, energy efficiency and conservation loan

program, and leveraging income tax on nonrenewable energy usage depending on the individuals or households income would help the transformation toward the usage of renewable energy.

Table 1 explained the behavioral theory that predicts low-income individuals to install PV. Three of the sections on low-income predictive behavior are divided into two parts: without intervention and with intervention. Generalization theory did not have any 'without intervention' column as generalization happens naturally once financial, knowledge, and social intervention are in place.

**Table 1.** Summary of 4 behavioral theory to support low-income individuals to install PV.

| Part | Behavioral Theory | Problem | Predictive Behavior Based on Behavioral Theory | Intervention |
|---|---|---|---|---|
| 1 | Matching Law theory (without intervention) | Financial and knowledge difficulty hinders low income individuals to invest on PV | Most low-income individuals are unable to invest on PV without the help of government of policy intervention | No intervention |
| | Matching Law theory (with intervention) | | Reinforcement such as financial support towards PV installation help nudge low-income behavior to invest on PV | Providing low-income individuals financial incentive to install PV nudges low-income individuals to invest on PV. |
| | | | | Policy programs providing low income with knowledge how they financially benefit from PV and how they seek for help to invest in PV would further increase their probability of low-income individuals to invest in PV. |
| | | | Public environmental policy providing financial subsidy to increase technological advancement and increase the supply of PV | Public policy providing financial subsidy to increase the economy and at the same time protect the environment. |
| 2 | Punishment (without intervention) | Feelings of uncertainty to invest in PV punished low-income individuals to invest in PV. | The feeling of not knowing how to invest in PV hinders low-income individuals from investing towards PV. | No intervention |
| | Punishment (with intervention) | | More knowledge on how to invest in PV replaces the punishing feeling of uncertainty to reinforcement to decrease energy bills. | Providing low-income individuals with education to how they can invest on PV removes the feeling of uncertainty and nudges low-income individuals to invest in PV. |

**Table 1.** *Cont.*

| Part | Behavioral Theory | Problem | Predictive Behavior Based on Behavioral Theory | Intervention |
|---|---|---|---|---|
| 1 and 2 | Punishment and reinforcement (with intervention) | Feelings of uncertainty to invest in PV punished low income individuals to invest in PV. | Social reinforcement is shown to encourage people to invest on PV. | Policy programs providing social events in which low income individuals can learn how to save and invest in PV. |
| | | | | Providing social reinforcers after they invest in PV are more likely to encourage people to invest on PV |
| | | | Financial reinforcement such as lottery helps encourage people to learn how they can invest in PV | Financial reinforcers to encourage individuals to learn how to obtain financial support to install PV and the benefit of PV |
| | | | Behavioral skill training are suggested to increase the probability of success in educators to provide reinforcers in natural ways | Training the educators how to provide reinforcers in a natural way. |
| 3 | Delay Discounting (without intervention) | Investing in Non RE is seen as a more small and immediate reward compared to investing on RE such as PV. | People who were unsure of the benefit of PV are more prone to choose small and immediate reward such as non-RE compared to installing PV. This is true even if they know PV would be more beneficial for them over the long term. | No intervention |
| | Delay discounting (with intervention) | | Providing financial reinforcement and making social groups that inform individuals on the benefit of PV. | With financial incentive and more knowledge about the benefit of PV low-income individuals are more to nudge to choose the more larger and delayed reward such as investing in PV |
| | | | Creating an incentive that aligns with individual reinforcement increases the probability of low-income individuals to invest on PV. | Involving low-income individuals among the decision-making policy programs process are likely to lead to policy change that is aligned with low-income reinforcement. |
| 4 | Generalization (with intervention) | | As people invest on PV, they are also likely to do more environmental friendly behavior. | To support low-income individuals to generalized more proenvironmental behavior, educators can align the benefits of PV with sentences which relates to the hedonistic, egoistic, altruistic, and biospheric values when educating low-income individuals on the benefit of PV. |

## 5. Implication, Conclusions, and Future Research Directions

This review provides a comprehensive behavioral theory to the adoption of RE behaviors to low-income individuals from a behavioral psychology perspective. The following research explains the barriers and proposes the intervention of low-income individuals to install PV. This research perspective is especially important as a big proportion of the world population is categorized as low-income individuals [94]. We believe low-income individuals will benefit greatly by adopting RE. By installing PV in the household, low-income

individuals have less burden to pay for electricity in the long term. The less financial burden allows low-income individuals to concentrate on focusing their finances in other areas such as food, health, and education [26]. Additionally, as equality increases, the government would have less energy burden to pay for the low-income individuals [95]. Low-income individuals can use their finances to focus on education, buying other goods and services, and the country economy can also be improved through tax [96]. As people invest more in PV, there are also direct benefits to the environment. There will be less global warming and increase in water level [5]. As a result of less global warming, there will be better water and air quality. All the advantages of low-income individuals to invest in PV is summarized in Table 2.

**Table 2.** Summarized the benefit of low-income individuals investing in PV.

| Entity | Benefit | Positive Side Effects |
|---|---|---|
| **Financial burden** | Lower financial burden for low-income individuals | More available finances for low income to spent for daily needs. |
| | | Improved quality of life as additional resources for education and leisure. |
| | | Government has less burden to support the low-income individuals finances. This furthermore leads to an increase economic system. |
| **Knowledge** | Improved knowledge about the ways to save money through electricity, ways to cut down or replaced electricity with PV in the long term. | Improved quality of life how to save electricity for the household. |
| | Improved knowledge of how they can invest in PV and save money for other expenses. With this knowledge, they can now consider the option to install PV | Improved planning for the future, rather than just planning to pay for the bills over short-term period. |
| **Quality of Environment** | Improved quality of the air and water environment | Improve individuals physical and psychological health as air and water quality improved. |
| | Less global warming and water level increase | More land for people to farm food, live, and more land for people to use for other renewable energy such as wind energy. |

This review has a few limitations. The first limitation is this study intervention is written based on theory. The experimental research has not been conducted yet to conclude that the theory works on renewable energy. However, as we have incorporated the theory and other applied research, this paper suggests that low-income individuals are more likely to install PV when there is an increase in government funding and the presence of social reinforcement. Additionally, this paper also suggests providing low-income individuals knowledge to attain the appropriate support to install PV, and understanding the benefits of PV enhances the probability of them installing PV. Based on this limitation, it is vital that future research tests these theories before applying the intervention to low-income individuals.

The second limitation is that this research looks at how to increase the adoption of low-income individuals in general. However, we know through various research that individuals have different barriers [65]. For example, there might be cultural differences regarding how low-income individuals overcome these barriers [65]. Different obstacles will therefore lead to a different intervention. Future research can look at how many percentages of low-income individuals can benefit from installing PV. It also can seem like if there are any other cultural differences in barriers of low-income individuals avoiding them invest in PV.

In conclusion, the transition of low-income individuals to install PV combat climate change and improve the health of the individuals involves changes in policy and human behavior. This research suggests providing incentives and social groups to learn about the benefit of PV help nudge low-income individuals to learn and install PV. The current study also suggests getting people involved in the decision-making process help to encourage low-income individuals to install PV. Helping low-income individuals invest in PV would also improve the country's economy. Not developing a program to help low-income individuals invest in PV would lead to the unsuccessful implementation of the RE goal to enhance global environmental quality. Furthermore, it would lead to a less healthy environment for the whole population. Hence, it is of great importance that policymakers and the government help these low-income individuals install PV for the benefit of the whole population.

**Author Contributions:** Conceptualization, U.R.; Data curation, S.-C.C.; Formal analysis, S.-C.C.; Investigation, F.A.R.; Methodology, F.A.R. and U.R.; Project administration, U.R.; Supervision, U.R.; Validation, S.-C.C.; Visualization, S.-C.C.; Writing—original draft, F.A.R. and U.R.; Writing—review & editing, F.A.R., S.-C.C. and U.R. All authors have read and agreed to the published version of the manuscript.

**Funding:** This research received no external funding.

**Institutional Review Board Statement:** Not applicable.

**Informed Consent Statement:** Not applicable.

**Data Availability Statement:** Not applicable.

**Acknowledgments:** The authors would like to thank University of Raharja, National Kaohsiung University; for their support in providing a place for this research through analysis of research systems.

**Conflicts of Interest:** The authors declare no conflict of interest.

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
