# Peer review of "Review of Behavioral Psychology in Transition to Solar Photovoltaics for Low-Income Individuals"

_sustainability, doi:10.3390/su14031537_

Round 1

Reviewer 1 Report

I have attached the comments file.

Author Response

The manuscript is well prepared and also looks attractive. My comments are minor and not critical. But could help to improve the manuscript.

  1. Response: Thanks for your positive encouragement 1. CO2 (Line no. 34) could be corrected to CO2
    Response: Thanks for the comment. In (Line No. 34), I have revised using the correct Math format.
  2. Figure 4 looks like a Table, and it is good to write Table 1?
    Response: Thanks for pointing out. In the middle of (Line No. 509 and 510), I have removed the Figure 4, andreplaced it with Table 1
  3. Figure 3 is an equation, and better not to give a Figure caption.
    Response : I have removed the figure caption in the equation and replace it (Line No. 398) with equation caption
  4. It could be good if the author could provide one more Table containing the information of benefits/social status changes/environmental advantages of low-income individuals after adopting the solar PV.
    Response : Thanks, for the recommendation. Between (Lines No. 555 and 556), I have added the table below in the implication sections
  5. One good research article came from Stefan Poier about the “Towards a psychology of solar energy”. It is good to go through it if you find something appropriate for your manuscript.
    Response : I have read this article thoroughly. I agree that this research would fit perfectly as supporting evidence of Steg et al research. I have added this in the introduction in (Line No. 165 - 175) as follows:
    a. Another internal factor to consider is the decision of multiple people in the households rather than just a single individual's decision to invest in PV. In many circumstances, the decision to invest in one household will need to be dependent upon the decision of multiple individuals [36] Knowing how multiple people in the households interact based on their different traits is important to predict whether they are going to invest in PV. This research looked at surveys concerning the behavior of investing PV for individual people and how people in a household with different traits interact with each other. This research suggests that the interaction of individuals in multiple households with different traits predictor affect their decision to invest in PV. Poier [36] research support Steg et al [35] research suggesting that aligning individual and multiple people values in households leads to a higher probability of them installing PV.
    Thank you for the recommendation.

Reviewer 2 Report

Dear Authors, 

I have general remarks.

Firstly, in the paper low income individuals should be describe by economic indicators. From methodological point of view I am not sure about general approach to low income individuals maybe you could show geographic differencies. Low income individual could be find in well developed countries please see (https://www.mdpi.com/1996-1073/13/23/6358) 

Secondly, why you zoom on PV? Could you explain this in the text.

Thridly, why are you skipping public policy? Is the any impact of public policy (in this context energy policy and social policy) on the raised issues? Please see (https://www.mdpi.com/1996-1073/14/16/4883). However mentioned text is on offshore wind but shows the impact of energy policy on RES development in economicly diversified country.

Last but not least in line 445 you propose some supportive actions however it seems to me that these recommendations are not the result of the paper. Maybe you could show wilder perspective of what countries could do to support this transformation. Of course there is also the place to act by other e.g. NGO, activists, etc.

Good job. 

Regards

Author Response

Firstly, in the paper low income individuals should be described by economic indicators. From a methodological pointof view I am not sure about general approach to low-income individuals maybe you could show geographic differences. Low income individual could be find in well developed countries please see (https://www.mdpi.com/1996-1073/13/23/6358)

Response: There are differences in how low income is defined by different research. In the manuscript (Line No. 79 – 82) I have provide the examples of low income from various of studies. According to countries too they have different definition to what is low income. As per your comment I have added the manuscript examples on how each country have different low income definitions. I argue in this research how it is very difficult to determine which population can benefit from the PV Renewable Energy. The scripts that I added is as below (can be found in Line No. 98 - 102):

Additionally, there is also different low-income definitions on different countries. In Finland low income is defined as household who have income lower than 60% of the median money income of all households across the country [32]. On the other hand, US low income is defined as the four people in the households that have income 50 - 80% of the state area [33].

I agree that evidence showed that low income individuals can be found in developed country as well. Thus in this research it only mention that PV investment programs and incentives were less seen in low income countries (rather than not seen). In this research we are trying to argue that PV installation is less found in low income countries. This is because low income individuals were less likely to invest on PV. We have edited the following manuscript to clarify this text. The edited text can be found in (Line No. 54 - 67):

Until today, PV is the most accessible type of RE installed by private entities and households for both high and low income individuals alike. The deployment of a smart grid coupled with advanced blockchain technologies accelerates the mass adoption of PV throughout the globe [15, 16]. Because of this characteristic, PV has been the fastest-growing type of RE compared to other types of RE [17]. However, the use of PV is less seen in low-income countries because low-income countries have more low-income individuals [13]. Low-income individuals have different barriers compared to high-income individuals, causing them to struggle to install PV. According to National Renewable Energy Laboratory [18], both low income and high income have similar personal obligations and personal interests to address environmental issues. However, although both groups have similar values, low-income individuals countries were seen to install PV less than high-income countries. Therefore, it is important to research the barriers of low-income individuals installing PV. Helping low-income individuals to install PV will help them financially and increase their quality of life [19].

Secondly, why you zoom on PV? Could you explain this in the text.
Response: We zoom on PV because PV are the easiest and most installed Renewable energy system in household. Household are important for them to feel safe and have higher quality of life. This is the reason why we focus our research in PV. The explanation in the manuscript is in Line No. 54 - 67.

Thridly, why are you skipping public policy? Is the any impact of public policy (in this context energy policy and social policy) on the raised issues? Please see (https://www.mdpi.com/1996-1073/14/16/4883). However mentioned text is on offshore wind but shows the impact of energy policy on RES development in economically diversified country.
Response : We have add sections on public policy (Line No. 296 - 307):

Palage et al. [68] study suggests that the government and public policy can encourage low-income individuals to invest in PV by providing financial subsidies to PV industry production, innovation technological advancement, and buyers. More supply and demand can be increased by providing financial subsidies to industry production and buyers in these sectors. By initiating this system, the economy can increase through the tax system, and at the same time environment can be protected. Public policy can also support technological advancement in the target to increase efficiency and lower the production cost of the PV. Based on matching law theory providing financial subsidies and incentives in this area will increase the sensitivity of the production sector to invest their time in PV. Additionally, it will also increase the sensitivity of the low-income individuals to buy PV. Based on matching law theory, increasing the price sensitivity towards all these sectors will improve the quality of the environment and the economy. (also updated in table 1).

We have already mentioned impact on the public policy in the text. There is an impact in the economy in which economy is going to have less burden to support the low income individuals. At the same time there will be an increase of the economy as education improved and people starts buying other goods and services through government revenue. In this research we use government and public policy interchangeably. This is because government and public policy actually worked together to make the economy better and at the same time improved the environment.

Government might be in charge of education and other tax systems. On the other hand, public policy is in charge how to implement the correct strategy to improve the economy and at the same time save the environment. So in a way they are interrelated. With this in the implication section I mention the following (Line No. 549 - 556):

Additionally, as equality increases, the government would have less energy burden to pay for the low-income individuals [95]. Low-income individuals can use their finances to focus on education, buying other goods and services, the country economy will also improved through tax [96]. As people invest more in PV, there will also be direct benefits to the environment. There will be less global warming, water level increase [5]. As a result of less global warming, there will be better water and air quality. Hence, all of these advantages will lead to an increase in population health.

Last but not least in line 445 you propose some supportive actions however it seems to me that these recommendations are not the result of the paper. Maybe you could show wilder perspective of what countries could do to support this transformation. Of course there is also the place to act by other e.g. NGO, activists, etc.
Response : As this research a review of the previous paper (not an experiment) the result of this paper would be recommendations. These limitation are mentioned in the limitation areas. 
As for how the countries could do better everything, as explained in the manuscript, that would depend on the financial incentive that the government put in place and also the exposure of the knowledge about the benefit of PV from different individuals (maybe can add into the text). Every culture and every countries depending on their available resources would have different ways on how different countries can support this transformation. As this paper focuses more in general, it did not mention specific ways that each country could do support the transformation. It only mention how the government in general can help the transformation towards PV for low income individuals. For the research to look at different countries, it would need to be survey, and experiment research, rather than a review (which is the research that I am currently doing). 
Thanks for your recommendation, We did add more information about how a country can support this transformation (the following text can be found in Line No. 530 - 539): 

Another way that countries can improve the system would be to initiate higher taxes for higher-income individuals using non-RE. The following tax money could then be used for the government to educate the low-income individuals on the benefits, how to install PV and how they can benefit in the long term. As individuals are sensitive to price, higher-income individuals are going to use less non-RE with the tax increase. Additionally, the following tax money can be used to subsidize technological PV advancement, PV suppliers, and incentivized PV buyers [68]. Solar for all, Energy efficiency and conservation loan program, and leveraging income tax on non-renewable energy usage depending on the individuals or households income would help the transformation towards the usage of renewable energy.

Reviewer 3 Report

Overall speaking, the topic of this study is very interesting and worth carefully examining. The article is well written, the logic is clear and straightforward, and the conclusions are meaningful and convincing. As a result, I would like to recommend the article to be accepted for publication in sustainability. Before formal acceptance, some minor revisions are needed to further improve the quality of the paper.

  1. the format of the paper needs to be revised according to the requirements of sustainability, including titles of sections, tables, references, etc.

2.Some references can be added to the discussion section such as

  1. doi: 10.3389/fpsyg.2021.721410
  2. doi: 10.24869/psyd.2020.420
  3. Bandyopadhyay SK, Goyal V, Dutta S. Problems and Solutions Due to Mental Anxiety of IT Professionals Work at Home during COVID-19. Psychiatr Danub. 2020 Autumn;32(3-4):604-605. PMID: 33373996.
  4. https://doi.org/10.1016/j.techsoc.2021.101844
  5. The limitations and possible future research directions should be interpreted at the end of the paper, so that in the future the researchers who are interested in this important topic could carry out follow-up studies.
  6. The establishment of the model needs to be supported by relevant literature.
  7. The choice of variables should be based on evidence, and it is best to add relevant references.
  8. More references need to be added in the mechanism analysis part.
  9. There is an error in the keyword in this article

10 The ref upgrading should also be supported by more references

  1. Many important references are missing in the calculation of resource mismatch.
  2. The discussion part of mediation effect needs more references.
  3. The conclusion can be refined.
  4. Policy suggestions can be written more specifically.

Author Response

1. The format of the paper needs to be revised according to the requirements of sustainability, including titles of sections, tables, references, etc.
Response: I have revised the titles of sections, tables, references, etc. according to your advice.

2. Some references can be added to the discussion section such as:
a) Doi: 10.3389/fpsyg.2021.721410
Response: Title: The Role of CSR Engagement in Customer-Company Identification and Behavioral Intention During the COVID-19 Pandemic.
This paper is irrelevant. Our research is on renewable energy, and the following research is on customer company relations during covid 19.

b) Doi: 10.24869/psyd.2020.420
Response: Title: Attitudes toward Complementary and Alternative Medicine, Beliefs in Afterlife and Religiosity among Psychiatrists, Psychologists, and Theologists.
We studied your recommendations. We focus our research is on renewable energy, and the following research is on the beliefs in the afterlife among psychiatrists, psychologists, and theologists.

c) Bandyopadhyay SK, Goyal V, Dutta S. Problems and Solutions Due to Mental Anxiety of IT Professionals Work at Home during COVID-19. Psychiatr Danub. 2020 Autumn;32(3-4):604-605. PMID: 33373996.
Response: Title: Problems and Solutions Due to Mental Anxiety of IT Professionals Work at Home during COVID-19.
Thanks for your recommendation. After we carefully studied, we didn't find a connection between our research on renewable energy, and the mental anxiety of IT professionals during Covid-19.

d) https://doi.org/10.1016/j.techsoc.2021.101844
Response: Thanks, and we carefully studied the suggestions. Our research is on how low-income individuals can install renewable energy. The following research is on the impact of the internet on green innovation (rather than actually getting people to invest in PV).

e) The limitations and possible future research directions should be interpreted at the end of the paper so that in the future the researchers who are interested in this important topic could carry out follow-up studies.
Response: Thanks for your constructive input. The limitation, future research directions have already been interpreted at the end of the paper. We have added clarity to the conclusion. For reference please see in (Line No. 557 - 566).

f) The establishment of the model needs to be supported by relevant literature.
Response: Thank you for your suggestions. We have added more supporting and relevant literature in Line No. 229-240.

g) The choice of variables should be based on evidence, and it is best to add relevant references.
Response: Thank you for your review. This research looked at past research with supporting evidence to study whether low-income individuals can benefit from PV Renewable Energy.

h) More references need to be added in the mechanism analysis part.
Response: Thank you, we have added more references in the result section.

i) There is an error in the keyword in this article
Response: Thank you. We have checked and corrected the errors.

j) The ref upgrading should also be supported by more references.
Response: Thank you for your constructive input. We have further revised to enhance the manuscript

k) Many important references are missing in the calculation of resource mismatch.
Response: Thank you. We looked back and studied any possible resource mismatch that could occur

l) The discussion part of the mediation effect needs more references.
Response: Thank you, Professor. We consider the mediation effect to be our limitation and our future research.

m) The conclusion can be refined.
Response: Thank you Professor for your constructive input. We have added more details to the conclusion part.

n) Policy suggestions can be written more specifically.
Response: Thank you for your valuable input. Policy suggestions have been added (Please see Line No. 296 - 307).
